# Flexural Damage of Honeycomb Paperboard—A Numerical and Experimental Study

**DOI:** 10.3390/ma13112601

**Published:** 2020-06-07

**Authors:** Leszek Czechowski, Wojciech Śmiechowicz, Gabriela Kmita-Fudalej, Włodzimierz Szewczyk

**Affiliations:** 1Department of Strength of Materials, Lodz University of Technology, 90-924 Lodz, Poland; 195985@edu.p.lodz.pl; 2Centre of Papermaking and Printing, Lodz University of Technology, 90-924 Lodz, Poland; gabriela.kmita-fudalej@dokt.p.lodz.pl (G.K.-F.); wlodzimierz.szewczyk@p.lodz.pl (W.S.)

**Keywords:** finite element method, honeycomb paperboard, flexural stiffness, failure criteria

## Abstract

This paper presents an experimental and numerical analysis using the finite element method (FEM) of the bending of honeycomb-core panel. Segments of honeycomb paperboard of several thicknesses were subjected to four-point flexure tests to determine their bending stiffness and maximum load. Several mechanical properties of orthotropic materials were taken into account to account for the experimental results. The numerical analysis of the damage prediction was conducted by using well-known failure criteria such as maximum stress, maximum strain and Tsai-Wu. The present study revealed how to model the honeycomb panel to obtain curves close to experimental ones. This approach can be useful for modelling more complex structures made of honeycomb paperboard. Moreover, thanks to the use of variously shaped cells in numerical models, i.e., the shape of a regular hexagon and models with a real shape of the core cell, results of the calculation were comparable with the results of the measurements. It turned out that the increase of maximum loads and rise in stiffness for studied samples were almost either linearly proportional or quadratically proportional as a function of the panel thickness, respectively. On the basis of failure criteria, slightly lower maximum loads were attained in a comparison to empiric maximum loads.

## 1. Introduction

Honeycomb paperboards made from organic and biodegradable raw material are extensively used in numerous industries. Their main advantages are low specific weight, high strength and stiffness in relation to their specific weight [1,2]. Other advantages of honeycomb paperboard are excellent energy absorption properties, insulation, thermal and acoustic properties [3]. Honeycomb structures can be observed in Nature in bones, bees’ honeycombs or stalks of grain. Paper honeycomb and its expansion production process was invented in 1901 by Hans Heilbrun [4]. In the late decade of the 1930s Lincoln manufactured paper honeycomb from Kraft paper which subsequently was used in building furniture. The formed sandwich panel consisted of a thin hardwood facing bonded to a thick paper honeycomb core. Recently, paper honeycomb cores and paperboards can be found in everyday used objects. They can be found in many objects around us, starting from box endings, pallets, inserts, fillings, fillings for doors, furniture, partition walls in construction and sandwich multilayer structures used in the aviation and automotive industries [5,6,7,8]. Due to the usage of cheap and recyclable materials it is possible to reduce the price of final products as well as limit the usage of natural resources. In the literature, there are numerous studies concerning both experimental and numerical studies of paper products. The most popular studies concern the analysis of corrugated paperboards and cardboard boxes. Fadiji et al. [9] investigated the influence of the geometrical configuration of vents on the mechanical strength of packaging. Similar studies were developed in [10,11,12], where among others. Zaheer et al. [13] analysed the strength of paperboard packages subjected to compression loads by using the finite element analysis. Patel et al. [14] investigated the local buckling and collapse of corrugated boards under biaxial stress. The stability and collapse of the corrugated boards were analysed in [15]. Bai et al. [16] analysed axial crushing of single wall corrugated paperboards. Wang [17] investigated the cushioning properties of paper honeycomb under impact. Chen and Yan [18] performed a study which aimed at determining the elastic modulus of sandwich panels fitted with Kraft paper honeycomb. Gu et al. [19] researched the in-plane uniaxial crushing behavior of honeycomb paperboard by considering three types of deformation modes. Paperboards filled with honeycomb subjected to compression were analysed by Wang et al. [20]. The behavior of paper honeycomb panels subjected to bending and compression load was studied in [21], where a simplified model of a honeycomb core with small displacements was applied to determine only the stiffness. Analyses of the strength of paper tubes are presented in [22,23,24]. On other hand, works concerning analyses of composite structures’ stability from a theoretical approach can be found in [25,26,27,28] and works including additional experimental results are [29,30,31,32,33,34,35,36,37,38]. Bolzon and Talassi [39] investigated the behaviour of anisotropic paperboard composites till collapse by using burst strength testers. Mentrasti et al. examined the behaviour of creased paperboard experimentally [40] and analytically [41]. Borgqvist et al. [42] examined the continuum model of paperboard material with a high degree of anisotropy. Other papers devoted to studies of honeycomb cardboards are [43,44,45,46]. Hua et al. [43] investigated the influence on edgewise compressive strength of two sandwich paperboards by using numerical and experimental approaches. The authors of [44] studied honeycomb paperboards under impact compression by using FEM and experiments. Mou et al. [45] analysed the in-plane bearing capacity of a honeycomb paperboard based on plastic deformations and plastic energy dissipation. Other papers devoted to studies of honeycomb cardboards are collected in [46,47].

Based on the aforementioned literature one can see that there are a few papers related to strength/stiffness of honeycomb paperboards but the topic still isn’t exhausted. The present study concerns experimental and numerical analyses of honeycomb paperboard subjected to 4-point flexural tests. Experimental tests were performed for three thicknesses of honeycomb paperboard in the two main directions of paperboard plane, i.e., the machine direction (MD) and the cross direction (CD) to assess the influence on stiffness and the maximum load. Moreover, different cell shapes were modelled to reflect the real (imperfect) shapes of honeycombs. Simulations based on FEM were performed for five thicknesses of honeycomb paperboard and for two main directions (MD and CD). Furthermore, the authors of present study took into consideration three failure criteria to predict the damage of panels. Contrary to the present work, this approach wasn’t used in other works referring to modeling the honeycomb paperboard on purpose to determine the maximum loads. Nonlinear simulations computed in substeps were performed in ANSYS^®^ version 2019 R2 for large displacements based on the Green-Lagrange equations [48]. In addition, various mechanical properties of orthotropic materials were assumed to adjust the numerical characteristics to the experimental curves. Finally, assessment of different heights of panels and validation of numerical models including strength of paperboards can be useful to conduct further simulations on more complicated structures. 

## 2. Materials and Methods

The object of analysis was paperboard filled with a honeycomb core. Such paperboards are produced in the form of panels as presented in Figure 1a. Cellular paperboard consists of two outer layers A and a honeycomb core B as is illustrated in Figure 1b.

Cellular cardboard is treated as an orthotropic body. This is due to two factors. First of all, it is caused by the nature of the core structure. Secondly, flat layers possess orthotropic material mechanical properties. In the plane of cellular paperboard, two main directions of orthotropy can be considered. The first one covers the direction of manufacturing (called machine direction or MD). The second (perpendicular) direction is called cross direction (CD). The main directions of the CD and MD of the paperboard are the same as paper used for the flat layers (CD_O_ and MD_O_—Figure 2). In the case of a paperboard core, the machine direction of the paper applied for the MD_R_ core is parallel to the height of the core. However, the cross direction CD_R_ is perpendicular to the height of the core.

The geometrical parameters of the cellular paperboard are described as: D—diameter of the circle inscribed in the regular hexagon (defined as the cell size), h—core height, H—paperboard thickness. Single thickness walls have the thickness of paper applied to manufacturing the cellular board core. However, walls glued to each other have double thicknesses.

### 2.1. Experimental Research

The experimental study was conducted in the Centre of Papermaking and Printing (Lodz University of Technology, Lodz, Poland). The research employed cellular cardboard in which both the core and the flat layers were made of Testliner type paper with a paperweight of 135 g/m^2^. Honeycomb paperboards with the same cell size (of *D* = 15 mm) and several thicknesses were tested. The following honeycomb paperboard thicknesses were considered: *H* = 8 mm, *H* = 18 mm and *H* = 28 mm. The 4−point bending tests of samples were carried out according to scheme depicted in Figure 3.

The distance between the supports and applied forces amounted to *L*_2_ = 200 mm and 2*L*_1_ + *L*_2_ = 400 mm, respectively. The samples used in test had the surface dimensions of 100 mm × 500 mm. Due to the orthotopic properties of cardboard, the measurements were carried out in the two main directions (MD and CD). The method used for cutting samples is depicted in Figure 4.

Before performing the bending tests, samples were dried at a temperature of 40 °C and subsequently they were conditioned according to standard PN-EN 20187:2000 (temperature 23 ± 1 °C and relative air humidity 50 ± 2%) [49]. The bending tests were carried out on Tensile Machine model Z010 (ZwickRoell, Ulm, Germany) equipped with a specialized tool (Figure 5a). The load range of the machine is from 0.1 N to 10 kN. The tool consists of four supports. Three of them (two upper ones and one lower) have two degrees of freedom (DoF) and the 4th one has only one DoF. During tests, the supports were moving at a velocity of 10 mm/min. The method used for placing the samples in the measuring grip is shown in Figure 5b.

Initial forces of 3 N and 10 N was applied for *H* = 8 mm, *H* = 18 mm and *H* = 28 mm, respectively. The tests were performed until full failure of the specimens. During the analysis, the force vs. deflection in the middle of the structure was measured. In paper materials, the bending stiffness (*BS*) is one of the basic strength indicators and refers to the width of the bent sample. In the case of the 4−point bending tests, the *BS* was calculated according to Equation (1).
(1)$$\mathrm{BS}=\frac{FL_{1}L_{2}^{2}}{16\mathrm{db}}$$
where *F* denotes the acting force, *L*_1_ and *L*_2_ are the distances between supports (see Figure 3), *d* means the deflection and *b* represents the width of the sample.

### 2.2. FE Model

Numerical analysis was performed for two shapes of honeycomb cell, because it was observed that real samples do not have a regular hexagon shape. Figure 6a shows the shape of an ideal hexagonal core cell. Figure 6b shows the modified shape of the core cell. Table 1 presents the values of the geometric parameters of the ideal cell (regular hexagon) and the dimensions of the assumed cell. In the case of regular hexagon *a* = *b* and length of the regular hexagon’s side can be determined from Equation (2) using the parameter *D* of paperboards as delivered by the manufacturer (Figure 2):(2)$$a=D/\sqrt{3}$$

The geometric parameters of the cells with the real shape were determined as average values taken from 20 measurements.

Numerical analysis was performed for a half of the panel (symmetry conditions were applied in the middle of the panel). The model was created by using shell 4node181 elements. According to the description [48], this element is suitable for analysing thin to moderately-thick shell structures. It is suitable for modelling composite shells or sandwich construction. While using shell elements it is necessary to assign thickness and material properties to the element. The walls of the honeycombs which are composed of two paperboards had double thickness (see Figure 2 and Figure 6a). The process of preparation of the numerical model started from creating the honeycomb core. First, a single honeycomb cell was drawn by gluing neighbouring areas. It was performed by transforming the work plane coordinate system. Subsequently, areas making up single cell of honeycomb were copied and glued together to create the whole core of a sample of 200 mm × 100 mm dimensions. Lastly, faces were created and connected to the core. The face subjected to compression was modelled with the line in the place of support in order to ease application of boundary conditions. Due to small gaps between the copied geometry (Figure 6a) it was necessary to apply geometrical tolerances. The honeycomb core had seven elements along its height. The element edge length was set to 2 mm. Such attributes provided an overall good mesh quality. In order to create an appropriate model, it was necessary to modify the local coordinate system of single finite elements because the default orientations of the shell elements did not agree with the orthotropy directions. Four local coordinate systems were introduced and the element coordinate systems were transferred to correspond with the orthotropy directions as seen in the real model. Then shell normals were modified to keep the proper orientation of elements (Figure 7a). To simulate 4-point bending it was necessary to apply proper boundary conditions. At the support two translations were constrained. In the middle of the panel, symmetry boundary conditions were applied by constraining two rotations and one translation. The load was applied at the shorter edge of the model as the translation of all nodes at the upper edge of the structure. Applied boundary conditions are presented in Figure 7b. The FE simulations were conducted for large displacements by using the Green-Lagrange formulation. The nonlinear computations performed in substeps were based on the Newton-Raphson algorithm. The number of substeps was set to be from 50 to 500. The number of iterations in each substep ranged from 10 to 500. The above described process was repeated for each of the examined paperboards. In numerical analysis paperboards with the following thicknesses were examined: Model 1—*H* = 8 mm, Model 2—*H* = 18 mm, Model 3—*H* = 22 mm, Model 4—*H* = 28 mm, and Model 5—*H* = 33 mm. In order to examine the influence of the honeycomb shape on the performance of the paperboard, it was necessary to create two models for each direction (MD and CD), one with the ideal honeycomb shape and second one with the real shape. Overall more than 60 numerical models were created.

In the FE model paper was modelled as linear-orthotropic in the elastic range [50]. The material properties were derived experimentally from data provided by the Lodz University of Technology Centre of Papermaking and Printing. Table 2 presents the mechanical properties of paper used in FEA. Variation 1 corresponds to the nominal values of material properties. Variation 2 and 3 correspond to properties decreased or increased by 20% with respect to nominal values, respectively. The ±20-percent difference was used because it usually happens that the actual values of the mechanical properties of the papers can differ by up to 20% from the nominal values given in the specifications.

Orthotropic material was modelled as linearly elastic (i.e., obeying Hooke’s law). In the present paper, progressive damage for orthotropic material hasn’t been taken into account but the analysis of failure was based on the failure criteria which allow determining the initiation of damage and estimating the maximum load (low estimation of load-carrying capacity). In this research three failure criteria are used: maximal-stress, maximal-strain [48] and Tsai-Wu [51]. The tensile and compressive strength for both directions of orthotropy has to be determined experimentally. Due to the low thickness of paper, it is almost impossible to determine the shear strength of paper experimentally. Therefore, Equation (3) [14] was used to approximate shear strength of paper. The strength parameters of the paper are presented in Table 3.
(3)$$S_{12}\approx \sqrt{C_{1}C_{2}}$$

## 3. Results and Discussion

In this section, the research results are presented and discussed. The results obtained experimentally (bending stiffness, maximum load *F*_max_ and force-deflection ratio Δ*F*/Δ*d* in the maximum load range (from 10% to 50%) are given in Table 4. 

The measurement results prove that as the thickness of the paperboard increases, the bending stiffness in both directions increases in the cardboard plane. In case of load-carrying capacity of samples under bending (in experiments), the maximum load rise corresponds almost proportionally to increase of paperboard thickness. It was also noticed that the determined bending stiffness (*BS*) and Δ*F*/Δ*d* were quadratically proportional to the increase of the panel thickness. This is the result of an increase in the moment of inertia of the bent cross-section due to the increase in the sample thickness. By comparing the experimental results of *BS* and Δ*F*/Δ*d* for MD and CD, a decrease by 35–45% was noted in the case of CD.

It has been observed that the location of the sample damage occurs in various places between the internal *L*_2_ supports. The reason of this effect is the presence of weakened places in the cardboard structure. Numerical models presented in this section are described according to the following scheme. Models with the close to the modified shape of the honeycomb cell contain letter *R* in the name of the model. The second number (number after the dot) describes the mechanical properties of the material (1—variation 1, 2—variation 2, 3—variation 3, see Table 2). Figure 8 presents the curves of load vs. deflection for Model 1 (*H* = 8 mm) of paperboards for MD. It can be easily seen that numerical models are shown to be stiffer than the real structures. Surely, the shape of the honeycomb cell influences the stiffness of the numerical model. For small panel deflections, the shape of the considered cell doesn’t matter.

However, with the increase of the deflection, the stiffness of models with modifed honeycomb shape decreases significantly. The load-deflection curves for Model 1 (*H* = 8 mm) of paperboard bent in the CD are presented in Figure 9. 

In this case the differences in stiffness caused by the shape of the honeycomb cell are smaller than in the previous case. However, contrary to samples loaded in the MD, the models close to the real shape of the core tend to indicate slightly higher stiffness than ones with the ideal hexagonal core. Models with the second variation of mechanical properties behave like the structures in the experiments (curves Experiment 1.1 CD and Experiment 1.2 CD run close to Model 1.2 CD and Model 1.2 CD R). Figure 10 shows the load-deflection curves for the model with *H* = 18 mm in the MD. Like the 1st model in MD, the differences in stiffness caused by the change of shape of the honeycomb cell are small for a certain value of deflection. Then, an abrupt divergence of the stiffness is observed. It turns out that Model 2.2 MD R is the closest to the experimental results.

Figure 11 compares the load deflection curves obtained for both cell of honeycomb for model 2 in CD. The curves obtained for models with the 2nd variation of mechanical properties (as in the previous case) are the closest to the experimental data. The curves obtained for both honeycomb cell shapes are close to each other, but the models with the modified (close to real) honeycomb cell shape seem to be stiffer.

Figure 12 shows load deflection curve obtained for the 4th analysed paperboard in the MD (Model 4). As in previous analyses, the behaviour of the numerical model is not the same as that of the real object. In a certain range of deflections there is a small influence of the honeycomb cell shape on the stiffness of the structure. At a certain value of deflection, the stiffness changes significantly. Moreover, the curves obtained for models with reduced mechanical properties are closer to the experimental characteristics but the trends of these curves at the beginning are slightly different.

The load-deflection curves for model 4 in CD are presented in Figure 13. The obtained characteristics of the load-deflection curves are similar to the previous ones. As it was noticed, load-deflection curves representing the behaviour of models with lowered mechanical properties are the closest to the experimental data. It is clear that models with a real honeycomb cell shape show higher stiffness, while models with the ideal honeycomb cell shape are closer to the experimental curves.

As presented in Figure 8, Figure 9, Figure 10, Figure 11, Figure 12 and Figure 13 in each case the load-deflection curves attained for models with variation 2 of mechanical properties seem to be closest to the experimental study results. It should also be noted that the model 2 in the direction of CD reflects very well the relationship between the force *F* and the deflection *d* obtained by experimental study. This means that it can be used to calculate the bending stiffness of cardboard, which is determined by the value of the Δ*F*/Δ*d* ratio (see Equation (1)). Therefore, the results presented for another variants are limited to variation 2 (with reduced mechanical properties). Figure 14 displays the comparison of charts for all considered models in the MD. The stiffness of the paperboard increases with the increase of the thickness of the paperboard. The influence of the shape of the honeycomb cell is insignificant in a certain range of deflections, but at a certain point the influence of the shape of the honeycomb cell became significant. Models with modified honeycomb cell shape appear to be less stiff.

Figure 15 compares the load-deflection curves for models in the CD. The general observations are similar to ones for models in the MD. However, in this case a reverse situation was observed because models with real honeycomb cell shape seemed to be stiffer than models with ideal the honeycomb cell shape.

The location of the failure propagation (place at which first signs of failure are observed) and the corresponding loads obtained for three failure criteria (Max-Stress denotes maximum stress criterion, Max-Strain represents the maximum strain criterion and Tsai-Wu means simply the Tsai-Wu criterion) are presented in Table 5. It can be easily seen that the failure load depends on the thickness of the paperboard. Moreover, the shape of the honeycomb cell has an essential impact on the results. The propagation of failure in models with ideal honeycomb cell shape is observed at higher loads than in models with modified honeycomb cell shape. The differences in the failure loads for models in the MD for different failure criteria are rather insignificant. In all cases, the first signs of failure (where stress states in these points were fulfilled based on the failure criterion) occurred in the outer layer subjected to compression (it happened mostly at the supports). After comparing the data presented in Table 5 with Figure 8, Figure 10 and Figure 12 it is visible that the failure forces attained for models with an ideal honeycomb cell shape are closer to the experimental study results. Moreover, the failure location indicated by the Tsai-Wu criterion corresponds to that observed in the experiments. Figure 16 presents the comparison of failure locations obtained for the Tsai-Wu criterion with the experimental results. In the case of numerical models, only half of the panel is presented. Figure 16a,b show that the failure occurred in the region near the support, while Figure 16c indicates the region near the center of modelled panel.

In Table 5, the failure loads based on the considered failure criteria are given. Referring to the failure criteria, the differences in maximum loads are comparable. However, by comparing the maximum loads based on numerical results and experimental ones, close correlations were observed (19 N for Model 1.2 MD and 15 N in the case of Model 1.2 MD R vs. 20.1 N for Experiment 1.1 MD and 19.7 N for Experiment 1.2 MD). In case of Model 2 MD, 44 N for Model 2.2 MD and 35 N for Model 2.2 MD R vs. 48.5 N for Experiment 2.1 MD, 50.2 N for Experiment 2.2 MD was noticed. In the case of Model 4 MD 72 N for Model 4.2 MD and 51 N for Model 4.2 MD R in a comparison to 71.2 N for Experiment 4.1 MD and 50.2 N for Experiment 4.2 MD was noted. Table 6 presents the failure loads and locations obtained for the investigated models in the CD. Contrary to the MD, the propagation of failure for panels in the CD is seen at higher loads for models with real honeycomb cell shapes. Comparison of the data presented in Table 6 with Figure 9, Figure 11 and Figure 13 leads to the conclusion that the failure loads obtained in the numerical analysis are smaller than the experimental ones. The failure loads obtained for models with real honeycomb cell shapes are closer to the ones obtained in the experimental study. In a comparison of failure loads determined numerically and experimentally for the CD, the following results were attained: 12 N for Model 1.2 CD and 13 N in case of Model 1.2 CD R vs. 20.2 N for Experiment 1.1 CD and 19.1 N for Experiment 1.2 CD. In the case of Model 2 for CD, the failure loads are the following: 26 N for Model 2.2 CD and 33 N for Model 2.2 CD R vs. 38.9 N for Experiment 2.1 CD, 38.0 N for Experiment 2.2 CD. In the case of Model 4 CD, 48 N for Model 4.2 CD and 59 N for Model 4.2 CD R were registered, in contrast to 65.3 N for Experiment 4.1 CD. In general, the numerical failure loads seemed to be smaller but it should be mentioned that the failure criteria allow indicating the initiation of failure (often called low estimation of load-carrying capacity) but not an accurate moment of the damage.

The Tsai-Wu criterion provides the lowest failure load for a particular cell type among all analysed failure criteria. Figure 17 illustrates the failure locations obtained for numerical models with real honeycomb cell shapes for the Tsai-Wu criterion with the experimental data. The failure locations indicated in Figure 17a,b are in agreement with the data presented in Table 6. In the case of Figure 17c, reaching the maximum stresses based on the given failure criterion (on the scale index greater than 1) was observed in the region near the support, but then the maximal value of the Tsai-Wu index was observed in the middle of the panel like in case of the experimental study. 

Table 7 presents maps of the failure index on the surface of the face subjected to compression for the three failure criteria and six consecutive load steps (forces from 5 N up to 30 N). A value on the scale smaller than 1 defines the safe region before the failure occurs. A value on the scale (index) greater than 1 means the limit stress state based on some failure criterion was exceeded. In these locations, damage propagations might develop. The greater value of the index on the scale means a greater possibility of local failure of the material. Based on maps, it can be observed that extremes of the Tsai-Wu and Max-Stress indexes are present in the support region but at some load value the extremes move from the support regions to the middle of the panel. It is worth mentioning that the transfer occurs at a force slightly higher than the force at which the ultimate stress state occurred (i.e., where the failure index is greater than 1). In the case of the Max-Strain criterion the failure location does not change. The failure index maps for the Tsai-Wu criterion differ slightly from the failure index maps for the Max-Stress and Max-Strain criteria because the Tsai-Wu criterion also contains in its formula [51] a factor representing the shear stress.

The distribution of failure indexes on faces subjected to compression for the three analysed failure criteria and five consecutive load steps for Model 1.2 CD R are presented in Table 8 (forces from 5 N up to 25 N). It can be observed that the Max-Stress criterion and Max-Strain criterion indicate almost the same locations of failure which for all steps are in regions close to the middle of the panel. In case of the Tsai-Wu failure index, its extreme is observed in the middle of the panel, but then it changes its position twice from the center of the panel to the region of the support, and then again to the middle of the panel.

The behavior of panels presented in Table 7 and Table 8 shows that in the case of real specimens the failure location should be expected to appear in multiple locations. Due to manufacturing issues, local damage during handling or sample preparation might happen so a real specimen will not fail in the place indicated as the one in which propagation of failure should begin. If the real specimen, due to the mentioned issues, happens to be stronger in the place at which it should fail first, or weaker in the place which is an extremum of the failure criteria at higher force (as presented in Table 7 and Table 8) then failure will occur in other locations. Based on the comparison of the results of measurements and calculations of the destructive load of cardboard in the MD bending test, it can be stated that the results of calculations using Model 2 do not differ much from the results obtained in the experiments.

## 4. Conclusions and Final Remarks

The paper presented an analysis of the bending of honeycomb paperboard panels. Numerical models were created for each direction and shape of the honeycomb cells. The numerical models were tested to validate the results of the experimental study. A study of the paperboards related to load-deflection curves due to four-point bending and failure criteria analysis was performed. Based on the obtained results, the following can be stated:Models with reduced paper mechanical properties provided curves close to the experimental curves.By comparing the maximum loads, the numerical prediction of failure loads gave lower results but it should be underlined that the failure criteria used only indicate (determine) the initiation of the damage in selected points based on an actual stress state (where the strength condition is fulfilled) but the failure criteria do not change in these locations the material properties. In experiments, local damage in panel might be possible what could lead to the load-carrying capacity of the bent panel being reached. Hence, discrepancies in failure loads based on both methods can amount to a few dozen percent, at most. It seems that these results can be acceptable.In general, among all considered failure criteria, the Tsai-Wu criterion indicated the failure propagation at slightly lower load.As far as the influence of the honeycomb cell on the results is concerned, models with the ideal cell shape were closer to the experimental study results for the MD.In all analysed cases the failure occurred on the face subjected to compression. Referring to both the experimental and numerical results, a pretty good agreement was attained. Moreover, the failure locations in the numerical model were observed in similar places to those where the failure of real specimens occurred.Model 2 reflects well the initial, straight line segment of the relationship between the deflection of the bending sample and the force causing the bending, which allows one to determine the bending stiffness of cardboard.Based on the experiments, an increase of the paperboard thickness causes an almost proportional increase of the maximum loads and a quadratic increase of stiffness (see Table 4).Validated models can be used for the creation of models with complex geometry.

## Figures and Tables

**Figure 1 materials-13-02601-f001:**
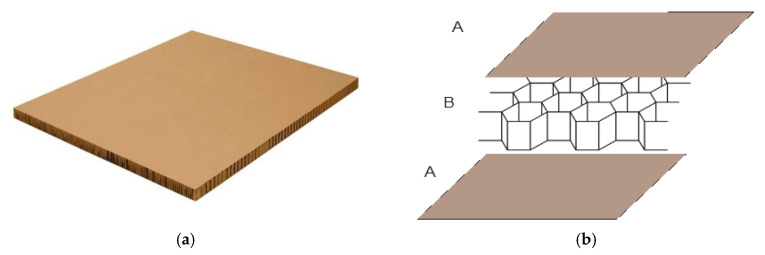
Analysed honeycomb paperboard (**a**) and construction of a cellular cardboard with a honeycomb core (**b**).

**Figure 2 materials-13-02601-f002:**
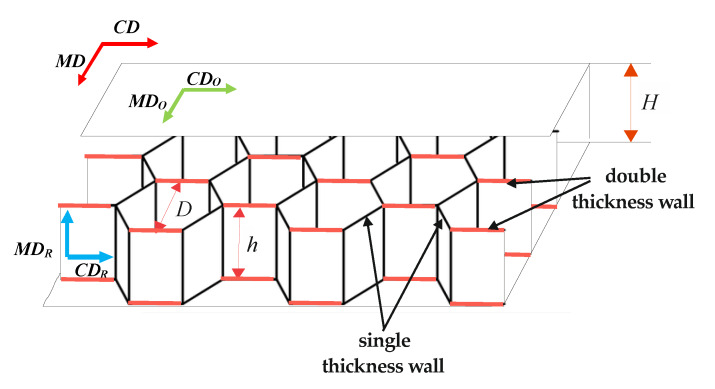
Construction of a cellular cardboard with a honeycomb core.

**Figure 3 materials-13-02601-f003:**
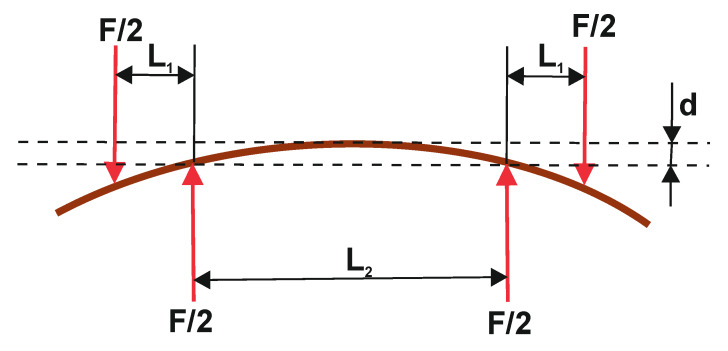
Loading scheme in 4-point bending test, F/2—forces acting in supports, L_1_, L_2_—distance between supports, d—deflection arrow.

**Figure 4 materials-13-02601-f004:**
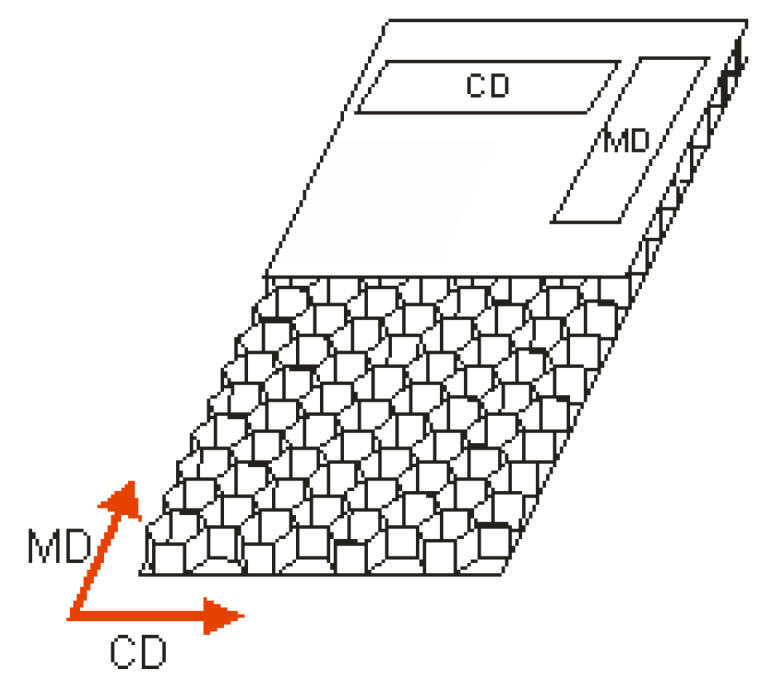
Method for cutting samples.

**Figure 5 materials-13-02601-f005:**
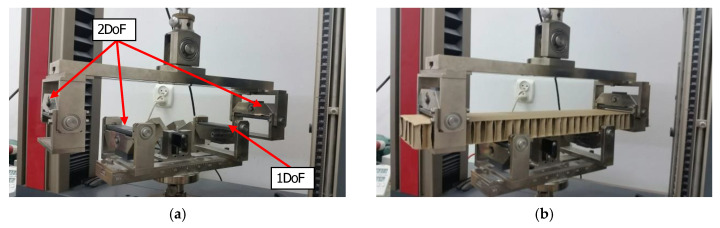
Tool for measuring bending stiffness (**a**) and a sample of the honeycomb paperboard placed in the tool (**b**).

**Figure 6 materials-13-02601-f006:**
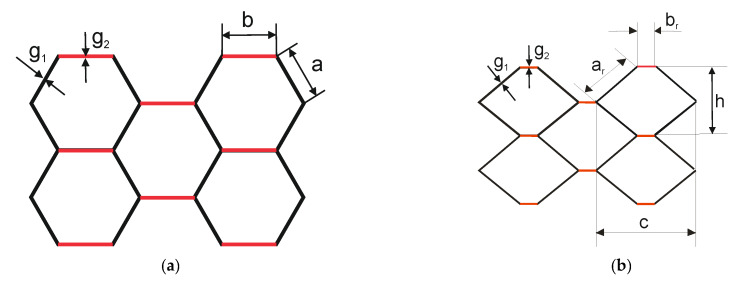
Perfect shape cell of the honeycomb core (**a**), modified shape cell (close to real shape) of the honeycomb core (**b**).

**Figure 7 materials-13-02601-f007:**
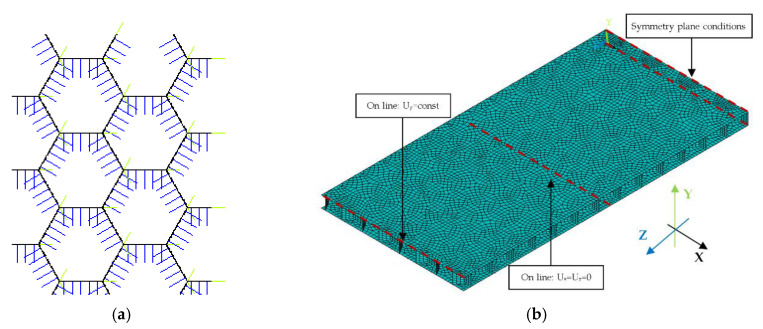
Orientation of local coordinate system in elements (**a**) and applied boundary conditions (**b**).

**Figure 8 materials-13-02601-f008:**
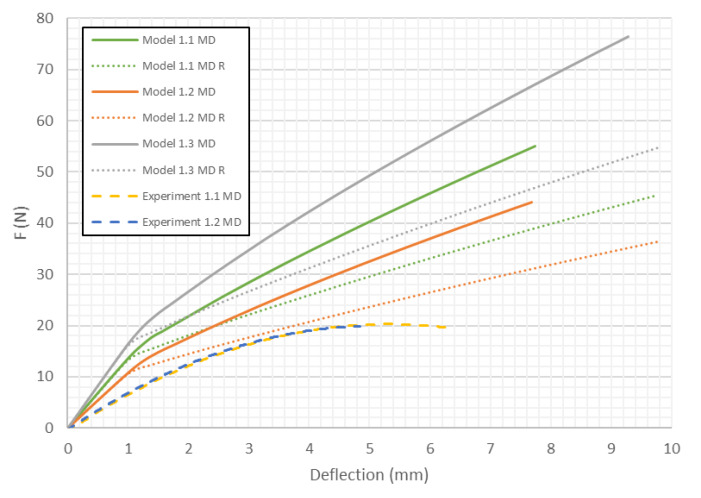
Deflection vs. force for Model 1 in MD.

**Figure 9 materials-13-02601-f009:**
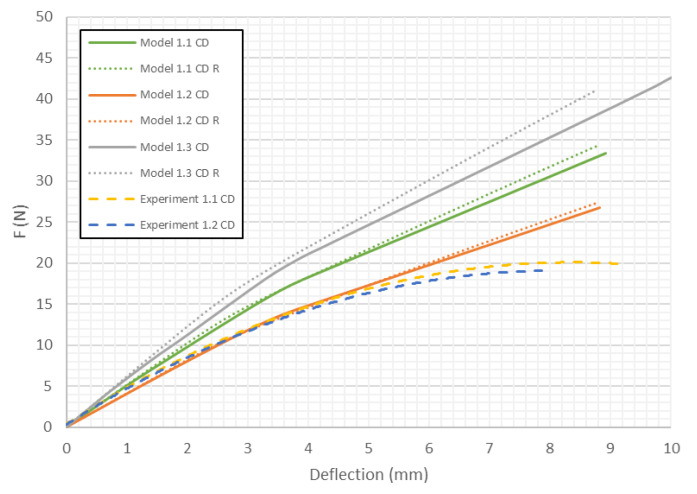
Deflection vs. force for Model 1 in CD.

**Figure 10 materials-13-02601-f010:**
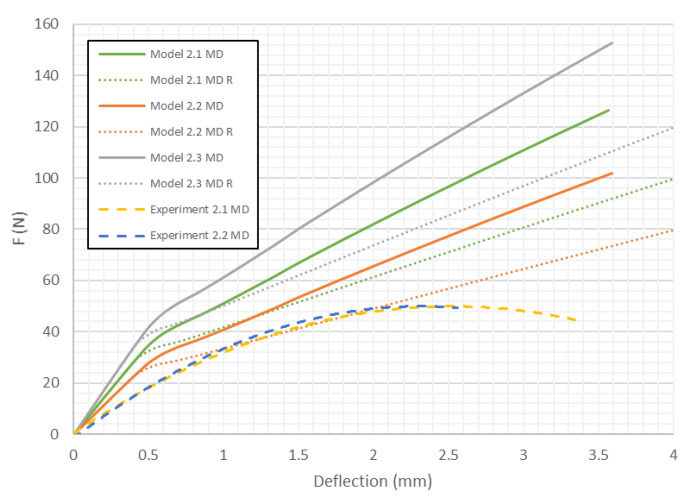
Deflection vs. force for Model 2 in MD.

**Figure 11 materials-13-02601-f011:**
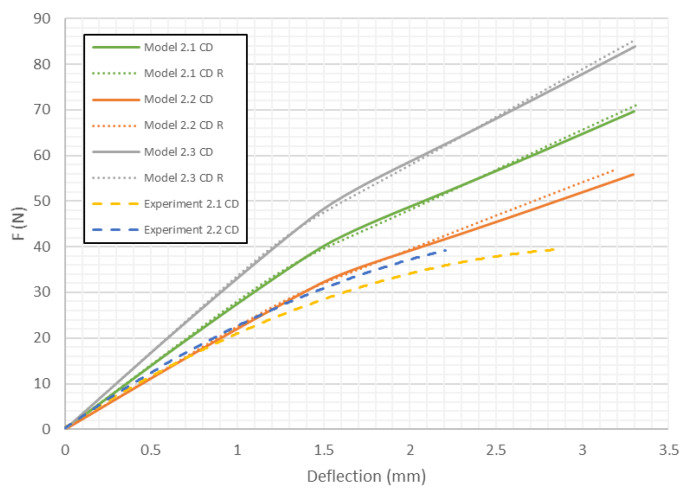
Deflection vs. force for Model 2 in CD.

**Figure 12 materials-13-02601-f012:**
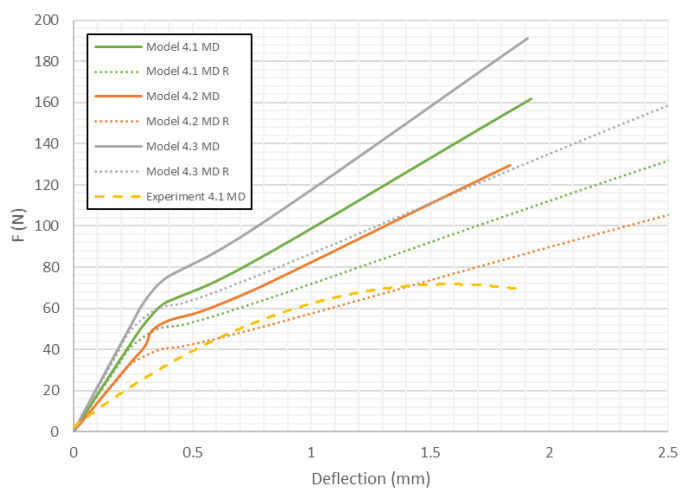
Deflection vs. force for Model 4 in MD.

**Figure 13 materials-13-02601-f013:**
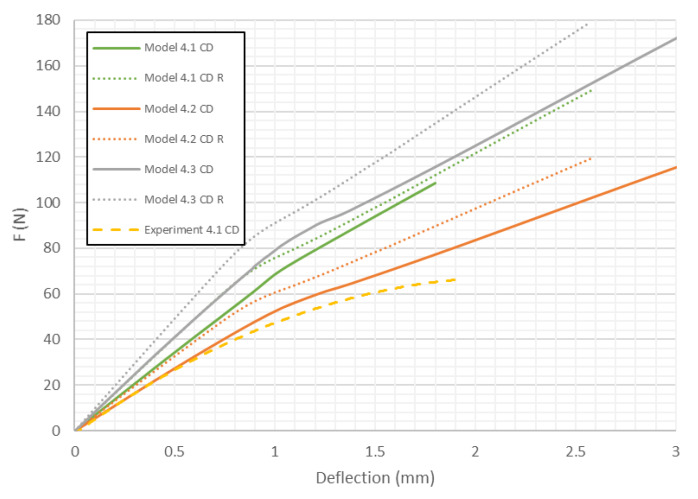
Deflection vs. force for Model 4 in CD.

**Figure 14 materials-13-02601-f014:**
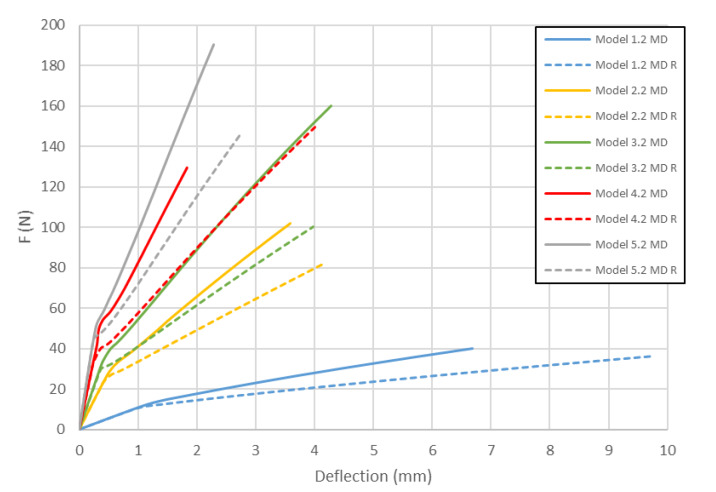
Comparison of curves of all considered cases in MD.

**Figure 15 materials-13-02601-f015:**
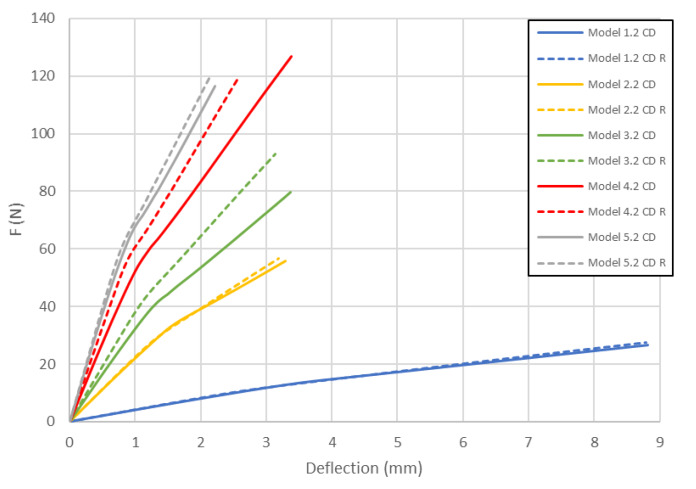
Comparison of curves of all considered cases in CD.

**Figure 16 materials-13-02601-f016:**
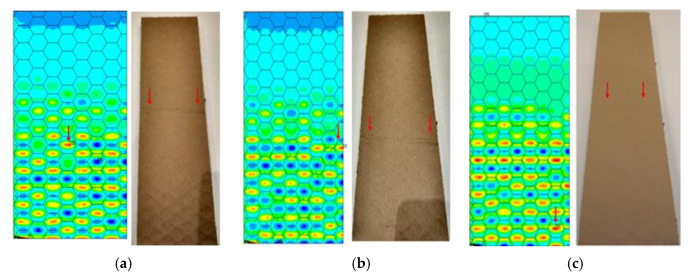
Comparison of failure places in MD: (**a**) *H* = 8 mm (**b**) *H* = 18 mm (**c**) *H* = 28 mm.

**Figure 17 materials-13-02601-f017:**
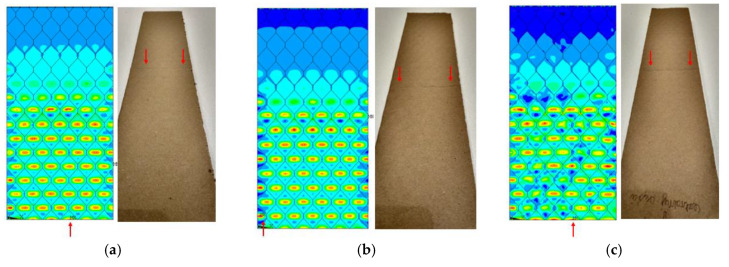
Comparison of failure places in CD direction: (**a**) *H* = 8 mm (**b**) *H* = 18 mm (**c**) *H* = 28 mm.

**Table 1 materials-13-02601-t001:** Ideal and real cell parameters.

Wall Length of a Single Ideal Cell	Wall Length of the Double Ideal Cell	Wall Length of a Single Real Cell	Wall Length of the Double Real Cell	Single Thickness Wall	Double Thickness Well	Real Cell Height	Real Cell Width
a (mm)	b (mm)	a_r_ (mm)	b_r_ (mm)	g_1_ (mm)	g_2_ (mm)	h (mm)	c (mm)
8.66	8.66	12.4	3.84	0.204	0.408	16.66	22.2

**Table 2 materials-13-02601-t002:** Applied mechanical properties of the material.

Variation	Young’s Modules in Cross Direction $E_{\mathrm{CD}}$ (GPa)	Young’s Modules in Machine Direction $E_{\mathrm{MD}}$ (GPa)	Shear Modulus G (GPa)	Poisson’s Ratio, $\nu_{\mathrm{MDCD}}$ (-)	Poisson’s Ratio $\nu_{\mathrm{CDMD}}$ (-)
1	1.846	5.323	0.934	0.315	0.109
2	1.476	4.258	0.747	0.315	0.109
3	2.215	6.387	1.120	0.315	0.109

**Table 3 materials-13-02601-t003:** Strength parameters of paper.

Strength Parameters	MPa
*T_1_*—tensile strength in direction MD	43
*T_2_*—tensile strength in direction CD	13
*C_1_*—Compressive strength in direction MD	16
*C_2_*—Compressive strength in direction CD	8
*S_12_*—Shear Strength	11

**Table 4 materials-13-02601-t004:** Results of the experiments.

Denotation of the Sample	Maximum Load *F*_max_ (N)	*BS* (Nm)	Δ*F*/Δ*d* (N/mm)
Experiment 1.1 MD	20.1	15.3	6.1
Experiment 1.2 MD	19.7	15.6	6.2
Experiment 1.1 CD	20.2	9.6	3.8
Experiment 1.2 CD	19.1	9.2	3.7
Experiment 2.1 MD	48.5	87.4	35.0
Experiment 2.2 MD	50.2	89.5	35.8
Experiment 2.1 CD	38.9	50.2	20.1
Experiment 2.2 CD	38.0	52.6	21.0
Experiment 4.1 MD	71.2	179.4	71.8
Experiment 4.1 CD	65.3	118.4	47.4

**Table 5 materials-13-02601-t005:** Comparison of failure loads and locations for models in MD.

Model	Cell Type	Tsai-Wu		Max-Stress		Max-Strain	
		Failure Load [N]	Failure Location	Failure Load [N]	Failure Location	Failure Load [N]	Failure Location
Model 1.2	MD	19	Near support	19	Near support	20	In the middle
	MD R	15	In the middle	15	In the middle	15	In the middle
Model 2.2	MD	44	Near support	44	Near support	44	Near support
	MD R	35	In the middle	35	In the middle	35	In the middle
Model 3.2	MD	54	Near support	55	In the middle	55	In the middle
	MD R	41	In the middle	41	In the middle	41	In the middle
Model 4.2	MD	72	In the middle	72	In the middle	72	In the middle
	MD R	51	In the middle	51	In the middle	51	In the middle
Model 5.2	MD	84	In the middle	84	In the middle	84	In the middle
	MD R	62	In the middle	62	In the middle	62	In the middle

**Table 6 materials-13-02601-t006:** Comparison of failure loads and places for models in CD.

Model	Cell Type	Tsai-Wu		Max-Stress		Max-Strain	
		Failure Load	Failure Location	Failure Load	Failure Location	FAILURE Load	Failure Location
Model 1.2	CD	12	Near support	12	Near support	12	Near support
	CD R	13	In the middle	13	In the middle	14	In the middle
Model 2.2	CD	26	Near support	26	Near support	28	Near support
	CD R	33	In the middle	33	In the middle	33	In the middle
Model 3.2	CD	32	Near support	34	Near support	34	Near support
	CD R	40	In the middle	40	In the middle	42	In the middle
Model 4.2	CD	48	In the middle	48	In the middle	48	In the middle
	CD R	59	Near support	64	Near support	64	Near support
Model 5.2	CD	49	Near support	49	Near support	49	Near support
	CD R	67	In the middle	67	In the middle	69	In the middle

**Table 7 materials-13-02601-t007:** Maps of the criteria index for Model 1.2 MD.

	F = 5 N	F = 10 N	F = 15 N	F = 20 N	F = 25 N	F = 30 N
Tsai-Wu	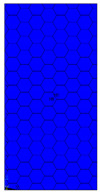		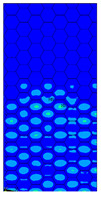	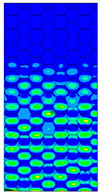	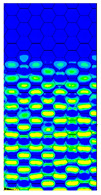	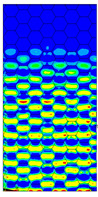
Max-Stress		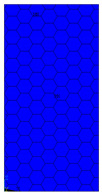	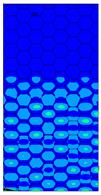	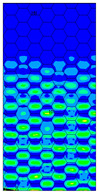	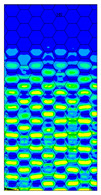	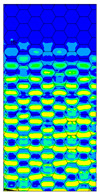
Max-Strain	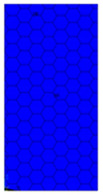	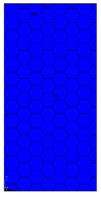	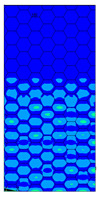	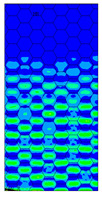	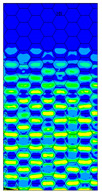	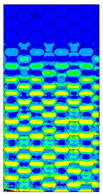
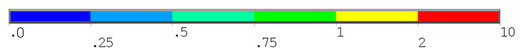

**Table 8 materials-13-02601-t008:** Maps of criteria index for Model 1.2 CD R.

	F = 5 N	F = 10 N	F = 15 N	F = 20 N	F = 25 N
Tsai-Wu	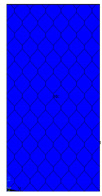	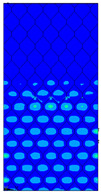	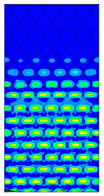	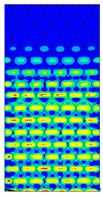	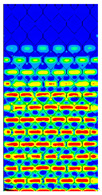
Max-Stress		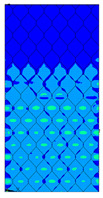	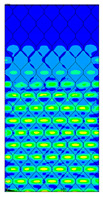	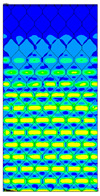	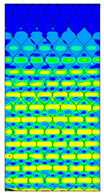
Max-Strain	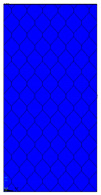	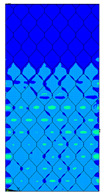	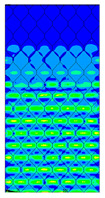	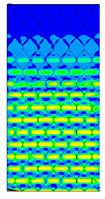	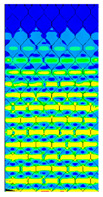
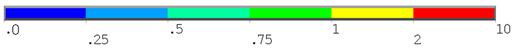

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
