# Peer review of "Flexural Damage of Honeycomb Paperboard—A Numerical and Experimental Study"

_materials, 2020, doi:10.3390/ma13112601_

Round 1

Reviewer 1 Report

The paper focuses on a comparative analysis between experimental and numerical analysis of honeycomb paperboard subjected to 4-point flexural tests. Experimental tests were performed for honeycomb paperboard samples with three different thicknesses. Detailed Finite Element Analysis simulations were then carried to assess the performance of paperboard samples and to compare with experimental results.

The subject of the paper is interesting and seems coherent with the scope and aim of the journal. The English is acceptable.

Some points must mandatorily be reviewed (major revision) before the paper can be taken in consideration for publication:

  1. Results are mainly given for FEM simulations. A section concerning a detailed discussion of the experimental results is missing. It is not easy to understand the results of simulations without a close look at the observed experimental behaviour (nonlinearities? damage onset and development? plasticity?). I suggest introducing a section reviewing in exhaustive manner all the experimental evidence.
  2. It is not clear whether the authors used geometrical nonlinearities in their FEM analyses. This feature could significantly affect the results of simulations in presence of large displacements. Please specify.
  3. It is not clear whether the authors used material nonlinearities (such as damage onset and development) in their FEM analyses. It seems that the employment of failure criteria is not supplemented by proper damage propagation. This feature could significantly affect the results of simulations in presence of evident nonlinearities. This issue should be properly discussed. Please specify.
  4. It is not clear whether the authors included manufacturing compaction phases (or residual thermal stress) in their FEM analyses. This feature could significantly affect the results of simulations, due to the heterogeneity of the tested material sample. Please specify.
  5. The aim of the study and its difference with respect to the established state of the art is not sufficiently stated in the introductory section. This should be amended.
  6. The discussion section is incomplete since adequate comparison with the observed experimental evidence is missing.
  7. The conclusion section does not properly reflect the findings of the work. As stated above, proper presentation of the experimental evidence is missing. Therefore, the comparison with FEM models is not conclusive.

Author Response

Dear Reviewer,

we’d like to thank for review of our manuscript.

The made changes in manuscript are highlighted on yellow.

Best Regards,

Leszek Czechowski

Wojciech Ĺšmiechowicz

Gabriela Kmita-Fudalej

WĹ‚odzimierz Szewczyk

Reviewer 2 Report

This paper presents a numerical and experimental study of flexural damage to the honeycomb paperboard.  The authors need to address the following comments:

  1. In abstract please provide a summary of the results
  2. What is the practical implementation of the current research? What are the noble contributions of the paper?
  3. What are the theories behind the finite element analysis?
  4. What are the reasons for taking different material properties and different types/sizes in consideration? The objective of the research is not clear.
  5. Did the authors consider a standard for the flexural test?
  6. The whole paper needs to be re-written in terms of technical discussion and by referring the recent publications
  7. What are the reasons for failure at the support and in the middle?
  8. Tables 6, and 7 contain redundant figures. How to calculate the criterion index?
  9. On page 6: please correct the table reference and Table numbering

Author Response

Dear Reviewer,

we’d like to thank for review of our manuscript.

The changes in manuscript are highlighted on yellow.

Best Regards,

Leszek Czechowski

Wojciech Ĺšmiechowicz

Gabriela Kmita-Fudalej

WĹ‚odzimierz Szewczyk

Round 2

Reviewer 1 Report

The quality of the paper has been enhanced.

I suggest the amendment of two minor points (minor revision) before publication:

  1. Page 6, line 185-186: “The FE simulations were conducted for large displacements by using Green-Lagrangian equations”. I propose to use “Green-Lagrange formulation” instead of “Green-Lagrangian equations”.
  2. Page 7, line 205: “Orthotropic material was modelled as linearly elastic (obeying Hook’s law)”. The phrase “Hook’s law” is wrong. This law makes reference to the English scientist Robert Hooke, not Hook, therefore it should be “Hooke’s law”.

Author Response

Dear Reviewer,

we’d like to thank once again for your insightful review of our manuscript.

The corrected text has been marked on yellow.

With Regards,

Leszek Czechowski

Wojciech Ĺšmiechowicz

Gabriela Kmita-Fudalej

WĹ‚odzimierz Szewczyk
